



# Technical note: Interpreting pH changes

Andrea J. Fassbender[1], James C. Orr[2], Andrew G. Dickson[3]

[1]Monterey Bay Aquarium Research Institute, 7700 Sandholdt Road, Moss Landing, CA 95039, USA
[2]LSCE/IPSL, Laboratoire des Sciences du Climat et de l'Environnement, CEA-CNRS-UVSQ, Gif-sur-Yvette,
France
[3]Scripps Institution of Oceanography, University of California, San Diego, 9500 Gilman Drive, La Jolla, CA 92093,
USA

*Correspondence to:* Andrea J. Fassbender (fassbender@mbari.org)

**Abstract.** The number and quality of ocean pH measurements has increased substantially over the past few decades such that trends, variability, and spatial patterns of change are now being evaluated. However, comparing pH changes across domains with different initial pH values can be misleading because a pH change reflects a relative change in the hydrogen ion concentration ($[H^+]$–expressed in mol $kg^{-1}$) rather than an absolute change in $[H^+]$. We recommend that $[H^+]$ be used in addition to pH when describing such changes and provide three examples illustrating why.

## 1. Introduction

In 1909, Danish biochemist Søren Peter Lauritz Sørensen proposed using a logarithmic scale to display the wide range of natural hydrogen ion concentrations ($C_H$–expressed in mol $L^{-1}$) in a more compact numerical form (Sørensen, 1909).

$$p_H = \log_{10}(1/C_H) . \tag{1}$$

The logarithmic scaling of hydrogen ion concentration derives from the Nernst equation, which relates the potential of an electrochemical cell to ion concentrations in solution, while the reciprocal form ensured predominantly positive values for $p_H$ in aqueous solutions (Sørensen, 1909). This definition was later amended to explicitly use the hydrogen ion activity ($a_H$) in aqueous solution (rather than the concentration) so as to take account of interionic forces when treating electromotive force data (Sørensen and Linderstrøm-Lang, 1924). This is the basis of the modern definition of pH (Buck et al., 2002):

$$pH = - \log_{10}(a_H) = - \log_{10}(m_H \gamma_H / m°) . \tag{2}$$

Here, $\gamma_H$ is the activity coefficient of $H^+$ (aq) at molality $m_H$, and $m°$ (1 mol (kg $H_2O$)$^{-1}$) is the standard molality. The negative logarithm was adopted by Sørensen and Linderstrom-Lang as a simpler way to express the original reciprocal. Summaries of modern pH scale development and refinement can be found elsewhere (Spitzer and Pratt, 2011). Still, since its inception, concerns have been raised about the inverse and logarithmic relationship between hydrogen ion concentration and pH being unintuitive (see Clark, 1922, especially p. 34) and the resulting increased likelihood of misinterpreted results. This prompted scientists to argue for alternatives to the pH scale, such as using specific acidity ($10^7$ minus the $[H^+]$) and its base 10 logarithm (i.e., 7 – pH; Clark et al., 1921; Wherry, 1919; Wherry and Adams, 1921). However, such efforts were unsuccessful and the Sørensen and Linderstrøm-Lang (1924) notional definition of pH, used early on as the basis for a conventional definition of pH and to assign values to pH standards (Bates and Guggenheim, 1960; Cohen et al., 2007; Covington et al., 1985; Hamer and Acree, 1939; McGlashan, 1970),



was ultimately adopted by the International Union of Pure and Applied Chemistry in 2002, thus defining pH explicitly through Eq. (2)  (Baucke, 2002; Buck et al., 2002; Cohen et al., 2007).

Within the field of marine science, several scales, all going under the name pH, have been commonly applied. These include one scale based on an operational approach that relies on calibration standards from the National Bureau of Standards (NBS; now the National Institute of Standards and Technology), and a variety of scales whose approaches all aim to realize pH as a "concentration" of hydrogen ion, usually expressed in mol kg$^{-1}$ (Bates, 1982; Dickson, 1984; Dickson et al., 2016; Marion et al., 2011; Waters and Millero, 2013). Such approaches were developed to simplify the use of acid-base equilibrium calculations in seawater media. The pH values presented here are on the total hydrogen ion  scale, the scale that is presently favored for measurement and reporting in observational oceanography (Dickson, 2010); however, the concern we illustrate here applies similarly to all marine science pH scales.

The immediate interest in assessing ocean pH changes is to help understand the consequences of rising atmospheric carbon dioxide ($CO_2$) levels, caused primarily by the combustion of fossil fuels, which result in a net transfer of $CO_2$ from the atmosphere to the ocean (Ciais et al., 2013; Friedlingstein et al., 2019). Once dissolved in the ocean, $CO_2$ reacts with water to form a weak acid that loses a hydrogen ion, which is largely neutralized through reaction with carbonate ion to form bicarbonate, causing the seawater $[H^+]$ to increase and the pH to decrease (Millero, 2007). This overall process is commonly referred to as ocean acidification (Caldeira and Wickett, 2003; Doney et al., 2009), and may have far reaching effects on marine life (Boyd et al., 2016; Doney et al., 2014; Hofmann et al., 2010; Kapsenberg and Cyronak, 2019; Kleypas et al., 2006; McNeil and Sasse, 2016; Orr et al., 2005) and on the rates of a variety of carbon cycle feedback processes within the ocean (e.g., Archer et al., 1998; Boudreau et al., 2018; Passow and Carlson, 2012; Revelle and Suess, 1957). As a result, it has become a priority in oceanography to monitor ocean pH and understand its natural and anthropogenic variations (Brewer, 2013).

Ocean pH is considered an Essential Ocean Variable (GOOS, 2019) and an Essential Climate Variable (GCOS, 2016) because it can be used to characterize ocean chemistry changes associated with anthropogenic carbon invasion and climate change, and it also meets the other desired criteria of measurement feasibility and cost-effectiveness. Distinct rates of persistent pH decline over decades have been observed across the global surface ocean at well-maintained time-series sites (e.g., Bates et al., 2014; Dore et al., 2009; González-Dávila et al., 2010; Munro et al., 2015; Olafsson et al., 2009; Sutton et al., 2014). Repeat hydrographic sections have also made it possible to characterize how ocean acidification is propagating into the ocean interior over decadal time scales (Byrne et al., 2010; Carter et al., 2017, 2019; Chen et al., 2017; Chu et al., 2016; Dore et al., 2009; Lauvset et al., 2020; Ríos et al., 2015; Watanabe et al., 2018; Williams et al., 2015; Woosley et al., 2016). Autonomous pH sensors capable of sustained observations (Aßmann et al., 2011; Johnson et al., 2016; Martz et al., 2015, 2010; Seidel et al., 2008) have begun to reveal the range and frequency of pH variations in open ocean and coastal waters (Bushinsky et al., 2019; Fassbender et al., 2018; Hofmann et al., 2011; Sutton et al., 2019; Takeshita et al., 2015; Williams et al., 2018). These observational efforts, in addition to numerical modeling studies (Bopp et al., 2013; Feely et al., 2009; Hauri et al., 2013; Jiang et al., 2019; Kwiatkowski et al., 2020; Orr et al., 2005; Resplandy et al., 2013; Steinacher et al., 2009), inform our understanding



of secular changes, patterns, and variability in ocean pH and guide research probing the sensitivities of marine organisms to changes in $CO_2$ system variables.

Since the beginning of the industrial era, it is estimated that ocean acidification has led to a global mean decline of ~0.1 in surface ocean pH (8.2 to 8.1), which corresponds to an $[H^+]$ increase of ~1.6 nmol kg$^{-1}$ (i.e., from 6.3 to 7.9 nmol kg$^{-1}$). It has not always been realized, however, that changes in pH reflect relative changes in $[H^+]$ rather than absolute changes. Most pH fluctuations in the ocean appear small, but for a given pH change ($\Delta$pH) the associated

$[H^+]$ change ($\Delta[H^+]$) varies, depending on the initial $[H^+]$ concentration. The relationship between these parameters can be derived as follows:

$$\Delta pH = pH_2 - pH_1 = - \log_{10}([H^+]_2) + \log_{10}([H^+]_1). \qquad (3)$$

and thus

$$-\Delta pH = \log_{10}\left(\frac{[H^+]_2}{[H^+]_1}\right). \qquad (4)$$

Here $[H^+]_2$ and $[H^+]_1$ represent the hydrogen ion concentrations corresponding to pH$_2$ and pH$_1$, respectively. The corresponding change in $[H^+]$ (i.e., $\Delta[H^+] = [H^+]_2 - [H^+]_1$) can then be shown to be

$$\Delta[H^+] = [H^+]_1 (10^{-\Delta pH} - 1). \qquad (5)$$

Equation (4) shows that changes in pH reflect a relative change in $[H^+]$, while Eq. (5) shows that the same pH change can equate to different $[H^+]$ changes (and thus reflect different chemical impacts of processes causing the pH change)

when implemented at different initial $[H^+]$ (or pH) values. For example, Fig. 1 shows that the same pH change results in a tenfold greater change in $[H^+]$ when starting at pH 7.4 instead of pH 8.4 (which will be true of any magnitude of pH change starting at these two values). The same point is made in a different manner by Kwiatkowski and Orr (2018). For studies evaluating trends and variability in pH, it is thus advantageous to also report results in terms of $[H^+]$ to make clear how the initial condition, $[H^+]_1$ in Eq. (5), influences the magnitude of the perturbation.

In the discussion, we provide three real-world examples that illustrate why reporting $[H^+]$ alongside pH can improve the clarity of studies that aim to evaluate changes in ocean chemistry. These examples include an evaluation of (1) modern sea surface trends, (2) the evolution of seasonal cycle amplitudes over the 21$^{st}$ century, and (3) changing interior ocean chemistry.

## 2.   Discussion

The first opportunity to improve clarity concerns comparison of pH changes between regions. Observed trends in sea surface pH typically fall between $-0.001$ yr$^{-1}$ and $-0.003$ yr$^{-1}$ (Bates et al., 2014; Byrne et al., 2010; Chu et al., 2016; Dore et al., 2009; González-Dávila et al., 2010; Ishii et al., 2011; Lauvset et al., 2015; Midorikawa et al., 2010; Munro

et al., 2015; Olafsson et al., 2009; Sutton et al., 2014; Takahashi et al., 2014). A critical piece of information that is often missing when such trends are compared (e.g., Table 3.2 of Rhein et al., 2013) is the initial pH value for each region. That information is key because regions with the same pH trend but different initial pH values will exhibit different $[H^+]$ trends over time (e.g., Fassbender et al., 2017). For example, Fig. 2a-b illustrates a scenario where two locations each experience a pH trend of $-0.0017$ yr$^{-1}$ (similar to that of the subtropics; *Bates et al.*, 2014) but have





different initial pH values: 7.9 and 8.1. As a result, there is a 58% greater change in [H$^+$] for the first relative to the second location. That is, when the change in pH is identical, the ratio between the two trends in [H$^+$] is equal to the ratio of the two initial [H$^+$] values. As a real-world example, Fig. 2c shows similar pH trends for two time series, the Equatorial Pacific (0 °N, 125 °W; Sutton et al., 2014) and Irminger Sea (64.3 °N, 28 °W; Bates et al., 2014), where the initial pH values differ (Table S1), causing the trends in [H$^+$] to differ. Yet, at another Equatorial Pacific site (0 °N, 155 °W; Sutton et al., 2014), there is a similar [H$^+$] trend to that of the Irminger Sea site because the initial pH differs. Recognizing that a change in pH represents a relative change in [H$^+$] and examining long-term trends in both parameters should improve interpretation of chemical changes across ocean domains.

The second opportunity to improve clarity concerns seasonal and diurnal variability of ocean $CO_2$ chemistry, both of which may condition the fitness and survival of organisms (Hales et al., 2017; Hofmann et al., 2011; Kapsenberg and Cyronak, 2019; McNeil and Sasse, 2016; Takeshita et al., 2015; Waldbusser and Salisbury, 2014). Identical peak-to-peak amplitudes of pH variations at locations having different annual mean pH implies different peak-to-peak amplitudes in [H$^+$]. While accounting for this concern affects interpretation of spatial patterns of pH variations, it appears even more critical when assessing how conditions evolve over time. For example, the seasonal amplitude of pH (A-pH) is expected to decrease while that of [H$^+$] (A-[H$^+$]) is expected to increase throughout much of the surface ocean over the 21st century under the RCP8.5 scenario (Kwiatkowski and Orr, 2018). This phenomenon arises because A-[H$^+$] increases relatively more slowly over time than the annual mean [H$^+$], (i.e., the numerator of Eq. (4) increases more slowly than its denominator). To illustrate how different these absolute and relative changes can be, in Fig. 3 let us compare simulated time series of pH and [H$^+$] sampled at the locations of the Kuroshio Extension Observatory (KEO; Table S2) and Drake Passage region north of the Antarctic Polar Front (DPN; Table S2) from the Geophysical Fluid Dynamics Laboratory's (GFDL) Earth System Model (ESM2M; Dunne et al., 2012, 2013) for the CMIP5 historical and RCP8.5 experiments (Riahi et al., 2011). Despite nearly identical decreases in A-pH at KEO (–0.0139) and DPN: (-0.0141) from the 1950s to 2090s, the corresponding change in A-[H$^+$] is not only positive at both sites but ten times greater at the former than the latter (1.70 versus 0.17 nmol kg$^{-1}$). Thus, it is desirable to assess A-[H$^+$] as well as A-pH.

The third opportunity to improve clarity concerns the interpretation of changes with depth, such as those between repeat hydrography line occupations or model time steps. Recently, the magnitude of chemical changes between repeat hydrographic sections have been inferred using various linear regression techniques (Carter et al., 2017, 2019; Chen et al., 2017; Chu et al., 2016; Lauvset et al., 2020; Williams et al., 2015; Woosley et al., 2016) and water mass characterization approaches (Resplandy et al., 2013; Ríos et al., 2015), with results often plotted in terms of ΔpH. Most ocean regions exhibit a larger range of pH in the upper 1000 m of the water column (~7.4-8.5; Lauvset et al., 2016; Lauvset et al., 2020) than across surface waters of the open ocean (~7.7-8.5; Lauvset et al., 2016; Fassbender et al., 2017). Because of these large vertical gradients in background pH, one cannot interpret the magnitude and pattern of the corresponding absolute chemical changes by studying only ΔpH. An example is given in Fig. 4 for a meridional section in the Pacific Ocean using the 2002-referenced Global Ocean Data Analysis Project mapped climatologies



(GLODAPv2.2016b; Lauvset et al., 2016). Despite there being larger changes in pH near the sea surface relative to the preindustrial period, changes in [H⁺] with depth indicate a different structure due to the heterogeneity of the background pH. Improved understanding of ongoing chemical changes comes from also studying Δ[H⁺], which reveals

aspects that studying ΔpH alone may conceal or overemphasize.

### 3.    Conclusions

When studying ocean acidification, the community often refers to changes in pH along with changes in other $CO_2$ system variables, such as $p$$CO_2$, total dissolved inorganic carbon, and the saturation state of seawater with respect to

aragonite. Yet the logarithmic scale of pH means that its changes are equivalent to relative changes in [H⁺], unlike for all other $CO_2$ system variables whose changes are not given on a log scale and are absolute. Unknowingly, many studies that have focused on ΔpH have described relative changes in [H⁺] presuming they were absolute. For absolute changes, one must actually compute Δ[H⁺]. We have illustrated this with three simple examples. Thus, when discussing changes in pH, it is recommended to show results as Δ[H⁺] as well as ΔpH, and when reporting pH data, it

is recommended to provide the reference conditions as well as the changes.

The Intergovernmental Panel on Climate Change (IPCC) defines ocean acidification as "*… a reduction in pH of the ocean over an extended period, typically decades or longer, caused primarily by the uptake of carbon dioxide ($CO_2$) from the atmosphere*" (Rhein et al., 2013; pp. 295; with a similar definition in Weyer, 2019; pp. 693). This apparent

emphasis on pH should be considered in the light of the challenges we have mentioned in interpreting pH changes in an ocean where background pH varies both in space and time. Do ocean regions with the same rate of pH decline really have the same rate of acidification, even if their initial conditions differ and hence their [H⁺] change varies? Does a greater pH change at the surface relative to the subsurface indicate greater acidification even if the change in hydrogen ion concentration is identical? Does a decline in the seasonal amplitude of pH imply benefits, given that the

opposite trend is projected for the seasonal amplitude of [H⁺]? Despite such concerns, the simplicity of the IPCC definition of ocean acidification continues to make it attractive. Whether or not it should be modified merits further discussion. More important is that the community move forward as a whole to go beyond reporting changes in pH alone, thereby avoiding the unwitting focus on relative rather than absolute changes in hydrogen ion concentration.

**Data availability**

All data used in this analysis are publicly accessible and the appropriate references, including doi, are provided.

**Author contribution**

AJF wrote the paper with contributions from all co-authors.


**Competing interests**

The authors declare that they have no conflict of interest.



**Acknowledgements**

AJF was supported by the David and Lucile Packard Foundation/MBARI, AGD by the US National Science
Foundation (OCE 1657799), and JCO by the French ANR Project SOBUMS (ANR-16-CE01-0014) and EU H2020
Project COMFORT (grant 820989).

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


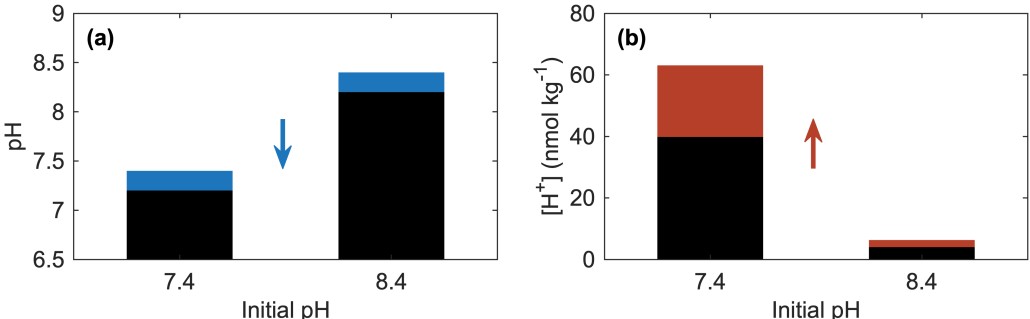

**Figure 1**. (**a**) A 0.2 unit decrease in pH (blue portion of bars) equates to (**b**) a 58% increase in [H$^+$] (red portion of
bars) for both initial pH values of 7.4 and 8.4. The absolute change in [H$^+$] depends on the initial conditions.

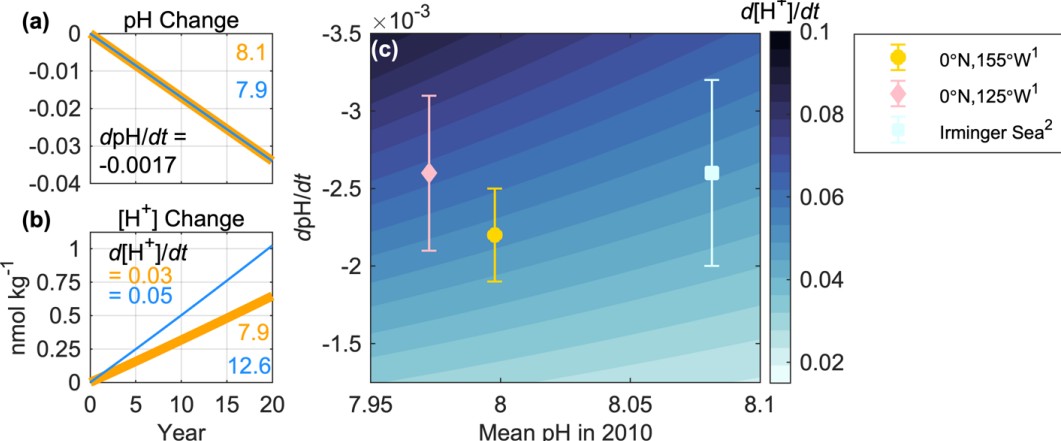

**Figure 2**. Change in sea surface (**a**) pH and (**b**) [H$^+$] over 20 years at two hypothetical locations. Changes are plotted
relative to the initial pH (8.1 and 7.9) and [H$^+$] (7.9 nmol kg$^{-1}$ and 12.6 nmol kg$^{-1}$) values noted in the figures. A fixed
pH trend ($d$pH/$dt$) of -0.0017 yr$^{-1}$ was imposed at both sites, resulting in [H$^+$] trends ($d$[H$^+$]/$dt$) of 0.03 nmol kg$^{-1}$ yr$^{-1}$
and 0.05 nmol kg$^{-1}$ yr$^{-1}$. (**c**) Contour plot showing linearized trends in [H$^+$] (nmol kg$^{-1}$ yr$^{-1}$) associated with mean (or
initial) pH values referenced to the year 2010 and the corresponding pH trends. The y axis is reversed so that larger
magnitude pH trends are near the top left corner. Symbols show observed surface ocean pH trends and uncertainties
(at in situ temperature) at select time-series sites, where legend superscripts refer to [1]Sutton et al., (2014) and [2]Bates
et al., (2014). Thirteen additional sites are included in Fig. S1. Details regarding the determination of mean pH values
referenced to the year 2010 are described in Text S1 and the values are presented in Table S1. The nonlinear
relationship between pH and [H$^+$] is not as apparent for small pH changes (such as those in panel **a**) as it is for larger
pH changes (such as the contours in **c**). Figure colormaps were made using cmocean (Thyng et al., 2016).

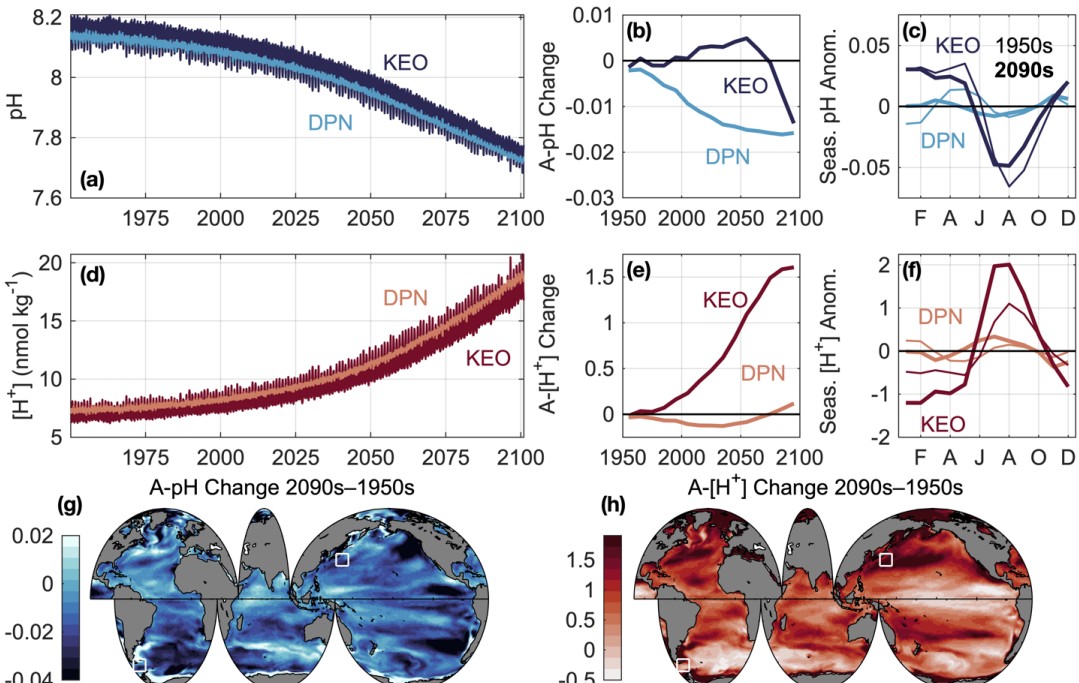

**Figure 3**. Data used in this figure come from the GFDL ESM2M model for the combined historical and RCP8.5 experiments. Time series of surface ocean (**a**) pH and (**d**) [H⁺] (nmol kg⁻¹) at model grid points corresponding to the Kuroshio Extension Observatory (KEO) location and the Drake Passage region north of the Antarctic Polar Front (DPN), similar to Region 1 in Munro et al., (2015). Surface ocean seasonal cycle amplitudes (A) were averaged for each decade and smoothed with a running mean filter using a four-element, sliding window. Shown are the decadal changes in (**b**) A-pH and (**e**) A-[H⁺] (nmol kg⁻¹) relative to the 1950s as well as the surface ocean (**c**) pH and (**f**) [H⁺] (nmol kg⁻¹) seasonal cycle anomalies at KEO and DPN during the 1950s (thin lines) and 2090s (thick lines). Global maps show the total change in (**g**) A-pH and (**h**) A-[H⁺] (nmol kg⁻¹) between the 1950s and 2090s. White boxes are centered at the KEO and DPN time-series sites. Simulated 1950s and 2090s annual mean pH and [H⁺] values as well as the 2090s minus 1950s change in A-pH and A-[H⁺] at these locations are listed in Table S2. 45 additional time-series locations are included in Table S2 and Fig. S2.
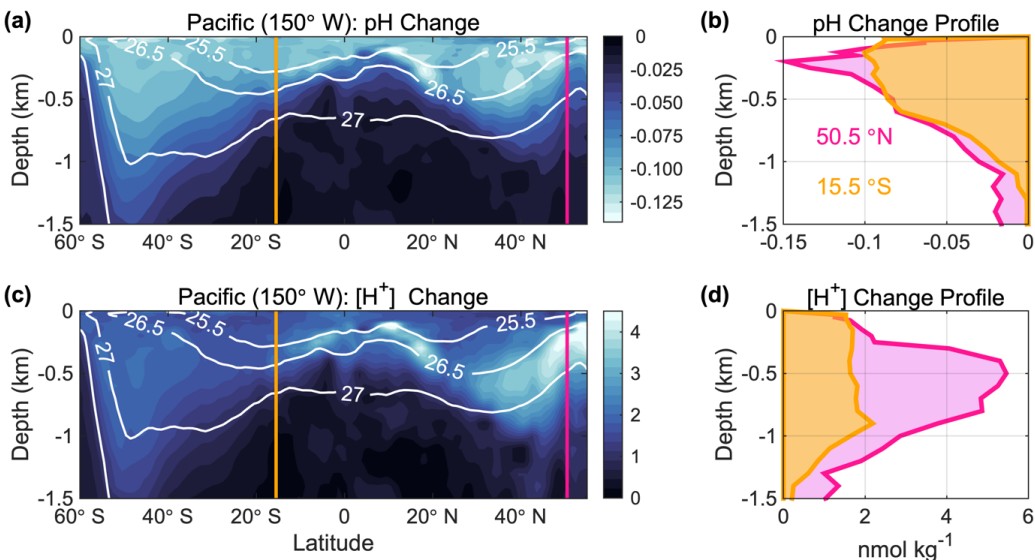

**Figure 4**. Changes in (**a**) pH and (**c**) [H$^+$] (nmol kg$^{-1}$) associated with anthropogenic carbon (C$_{anth}$) accumulation in
the upper 1.5 km of the ocean along a meridional transect (150º W) in the Pacific Ocean from 60º S to 55º N. These
values were estimated using GLODAPv2.2016b climatology data, which is referenced to the year 2002 (Lauvset et
al., 2016), by subtracting the estimated C$_{anth}$ from dissolved inorganic carbon (DIC) and recalculating pH from the
modified DIC values along with total alkalinity, silicate, phosphate, temperature, and salinity climatology data. Plotted
values are derived from the differences between the pH climatology and recalculated pH values that roughly reflect
preindustrial values. This simple approach neglects pH changes caused by processes other than C$_{anth}$ accumulation and
is used merely to display the concept of interest. Potential density contours are overlaid in white. Vertical profiles of
(**b**) pH change and (**d**) [H$^+$] change at the gold and pink lines shown in panels **a** and **c**. Calculations were performed
using the MATLAB program CO2SYS version 1.1 (van Heuven et al., 2011; Lewis and Wallace, 1998) and applying
the equilibrium constants of Lueker et al., (2000) and Dickson, (1990) and the boron-to-chlorinity ratio of Uppström,
(1974), following the recommendations of Orr et al., (2015) .
