# Peer review of "Technical note: Interpreting pH changes"

_Biogeosciences, 2020_

## Short Comment (SC1) · 17 Oct 2020

First, I think this paper will be an excellent addition to the fields of marine pH and ocean acidification (OA) research. I wholeheartedly agree with the notion to report proton or hydrogen ion concentrations which I have encouraged this with one of my own papers and I hope this paper will stimulate further discussion among scientists and environmental managers in regard the reporting pH and proton concentrations in future OA studies undertaken across a variety of disciplines. However, I think the scope of this technical note is too narrow given it predominantly focuses on measurements collected in open ocean environments whereas reporting proton concentration and pH is just as important in nearshore estuarine and coastal and freshwater systems. Details below.

[Figure]

Abstract - Looks good with no further comments.

Introduction - The authors provide a good overview and summary of the chemical history of pH dating back to the work of Sorenson in the early 19th century up to present day with the development of the three concentration scales and widespread development autonomous chemical sensors capable of measuring pH while sufficiently explaining why pH is important, its use in marine carbonate chemistry and OA, and the relationship/conversion of/between pH and proton concentrations. Literature cited is sufficient.

One suggestion for improvement though is the need to address comparing trends in marine pH across programs, years, and sites that measure pH on different scales and different methods. The need to establish inter-comparability between different pH datasets is necessary before calculating proton concentrations from pH if pH were measured on different scales across years, sites, or programs because the differences between scales would lead to systematic errors in the calculated proton concentrations and its relative changes further complicating the use and interpretation of those data. The manuscript mentions that all pH data are reported on the total scale. Notwithstanding, a couple of sentences explaining this may help scientists and environmental managers with little prior knowledge of the chemical history of pH and pH metrology as I think this type of work holds potential utility for folks working in regulatory environments that do not necessarily have their a finger on the pulse of this particular field.

Discussion

This bulk of this section is conservatively a follow-up to Fassbender et al. (2017) (Lines 281-282) which lays the groundwork for the real-world examples that illustrate why report proton concentrations alongside pH can improve the clarity of studies that aim to evaluate changes in ocean chemistry.

The above mentioned paper explains the need to evaluate non-significant long-term decreasing trends in marine pH measured by ocean time-series programs against the

same trends in proton concentrations which are likely to be better metrics given different starting/initial pH values in those systems. I think this section can easily expanded to include a paragraph or two that covers the same dynamic but for nearshore estuarine and coastal systems that experience a range of processes that modulate changes in pH, acidification/basification, and marine carbonate chemistry. Carstensen et al. (2019) pulled pH from 83 coastal ecosystems and calculated annual rates of change in pH between -0.23 and +0.23 pH/year which are consistently an order of magnitude greater than those of ocean time-series programs estimated by Bates et al. (2014) (Lines 194-196). I would recommend the authors draw a small subsample of data presented in that paper to illustrate why reporting proton concentrations and pH can hold just as much if not more utility in nearshore systems given how much larger rates of pH change are there, the inherent variability of background pH conditions and the number, timescale, frequency, and type of processes that impact pH in these systems. Alternatively, data from a paper like Lowe et al. (2019) may work as well since it pulled data from 83 sites in Puget Sound and Washington State's Coastal Estuaries (both in the USA) that experienced a broad spectrum of pH trends over time also greater than what absorption of anthropogenic CO2 alone can explain.

The second example is sufficiently explained but I would recommend that the authors that state that marine organisms respond to proton concentrations or acidity rather pH ti drive their point home (use refs for this cited in Fassbender et al. (2017)).

The third example examining changes across depth is also critically needed moving forward. I would further postulate that the change in proton concentration with depth or proton concentration depth gradients could provide a valuable complement to changes of saturation state with depth/depth gradients as well.

For a fourth example, I would strongly recommend at least including a paragraph on the use a proton concentrations as a means to view acidification in nearshore estuarine and coastal systems through the lens of proton cycling and proton fluxes both within individual systems and across/between interconnected systems. Just as CO2 can outgas in transit between the head of a large estuary like Delaware Bay and the Atlantic Ocean, protons are produced and consumed by the range of processes that modulate acidification like dilution by freshwater, photosynthesis, and respiration and in transit as well. It is the net result of these proton consuming- and producing-processes (i.e., proton cycling) that ultimately results in acidification or basification in nearshore estuarine and coastal systems. I understand this is a relatively new application and method for OA studies but it is a simple and straightforward forward one. I have attached a paper (Pettay et. al (2020)) outlining the proof of concept for this application that was done using data from the Murderkill Estuary-Delaware Bay System in Delaware, USA. Essentially, once you convert pH into a proton concentration it can be used and treated just as any other dissolved constituent in aquatic systems (e.g., DIC or TA) would be and from this work we conclude that the Murderkill Estuary is acting as a proton sink and sequestering protons from Delaware Bay on monthly and seasonal timescales and locally buffers portions of the Bay. Such dynamics may apply more broadly in Delaware Bay and other systems around world.

Since this applications requires additional environmental data and work beyond the simple calculations outlined in equations 3 and 4 in the manuscript, it may lie beyond the scope and intended purpose of this manuscript. Two follow-up analyses/papers are currently in prep that - (1) Look at multiple years of proton flux data in the same systems to examine annual and interannual trends in the proton source-sink dynamics and (2) Linking trends in proton concentrations and fluxes to other marine carbonate system parameters and nutrients to examine interactions between acidification and eutrophication in the same system. Works remain in progress but the initial results are promising. I definitely think will turn into a useful application of proton concentrations for nearshore OA studies but may lie beyond the scope of this manuscript and detract from the main points this manuscript already makes.

Conclusions - Sufficient for the information/discussion provided in the first draft of the manuscript. Please modify accordingly if needed if any of the suggested addi-

tions/revisions are incorporated into the final version.

References (not already included in the manuscript) -

Carstensen, J., & Duarte, C. M. (2019). Drivers of pH variability in coastal ecosystems. Environmental science & technology, 53(8), 4020-4029.

Lowe, A. T., Bos, J., & Ruesink, J. (2019). Ecosystem metabolism drives pH variability and modulates long-term ocean acidification in the Northeast Pacific coastal ocean. Scientific reports, 9(1), 1-11.

Pettay, D. T., Gonski, S. F., Cai, W. J., Sommerfield, C. K., & Ullman, W. J. (2020). The ebb and flow of protons: A novel approach for the assessment of estuarine and coastal acidification. Estuarine, Coastal and Shelf Science, 236, 106627. (also attached)

Please also note the supplement to this comment:
https://bg.copernicus.org/preprints/bg-2020-348/bg-2020-348-SC1-supplement.pdf

―――――――――――――――――――

---

## Referee Comment (RC1) · Anonymous Referee #1 · 2 Nov 2020

The manuscript is a brief explanation and discussion of why calculating and discussing trends in [H+] is necessary and important in order to correctly interpret pH changes across different oceanic regions and depths. As it is a technical note the manuscript is short, and there is little in-depth description and discussion. However, the referencing is more than adequate to guide the interested reader further into the topic. In that respect the introduction provides a very nice historical overview of pH. The manuscript is well-written and very nicely presented. The figures all have high quality.

I have four minor comments, which I think it would be useful for the authors to address:
1. How pH is measured has changed dramatically over the years, and is still changing. It would be interesting to include a brief description of what is actually measured when using spectrophotometric methods versus ion-sensitive field-effect transistors. I realize

the authors may think this beyond the scope of this work, but in terms of interpreting pH changes I believe this issue is becoming more and more important.

2. The manuscript does not mention biological responses. Probably this is with very good reason since it goes beyond interpreting pH changes. However, research into ocean acidification is well aware that this issue is much more than just pH and always make sure to also include changes in carbonate ion, or the saturation states of calcium carbonate minerals. It would be worthwhile to acknowledge this.

3. In the conclusions the authors make the statement "Unknowingly, many studies that have focused on delta-pH have described relative changes in [H+] presuming they were absolute." The statement has no references and I am not sure I believe it to be true. The statement makes it sound as if the general ocean carbonate chemistry community is unaware that an absolute change in pH represents a relative change in [H+]. This is not my understanding at all. While I agree that there has been too much focus on changes in pH alone and that also discussing [H+] is necessary, I doubt this is due to ignorance. At the very least the statement needs references as examples of this.

4. I would be worth mentioning (briefly) that changes in ocean pH, and [H+], occur as a result of perturbations to the carbonate chemistry buffer system. Something changes the balance between carbonate ion and bicarbonate ion and this results in a perturbation of pH. Not the other way around. Related to this a bit more discussion about the differences between coastal and open ocean would be worthwhile.

———————————————————

---

## Referee Comment (RC2) · Anonymous Referee #2 · 16 Nov 2020

Review of Fassbender et al., Interpretation of pH changes The consideration presented in this "Technical Note" are interesting, important and a useful guide for all those who are involved in any research related to acidification. Although no new physical-chemical concepts are presented, I am pretty sure that many studies concerning acidification and related biogeochemical issues were not aware of the different meaning of $\Delta$pH and $\Delta$[H+] and of the implications for biogeochemical processes. The three case studies illustrating the necessity for distinguishing between $\Delta$pH and $\Delta$[H+] are convincing. Why not publishing the manuscript as a regular research paper, together with some parts of the "Supplementary Material" and a more detailed interpretation of the calculations/Figures in the main text and not in a compressed form as Figure captions? In other words: I recommend this manuscript for publication. Some minor comments:

[Figure]

Line 477, Fig 2: better: $\Delta$pH and $\Delta$[H+]; Line 487, Fig. 3: Please define the "anomaly of the seasonality"; References: You have 6 pages for references, but less than 5 pages for the main text. Is that a reasonable proportion? Perhaps it is more informative if you confine the references to a few key papers.

———————————————

---

## Author Comment (AC1) · 15 Dec 2020

Thank you for your interest in the manuscript and for providing constructive feedback.

Below we address each comment from your review in order of appearance.

We agree that challenges associated with merging ocean pH observations remain a barrier to creating accurate, long-term pH records. The difficulties result from the problems inherent in clarifying the likely uncertainties in pH measurements made by different individuals at different times or places and are exacerbated by changes in measurement technique and/or calibration approach (pH scale) between research groups. However, we do not think this short Technical Note is the ideal place in which to address such complex challenges. On Line 43 we note that pH is presented on the total hydrogen ion scale throughout the article and thereafter focus on issues associated with the presentation and interpretation of pH change data that are already on a common pH scale.

Thank you for bringing the Carstensen and Duarte, 2019 publication to our attention. We have elected not to add a fourth, coastal example to the manuscript, as pH changes will always represent relative changes in [H+] no matter the location. We also feel that lengthening the manuscript and providing more than one example for any of the three cases presented would make it less accessible. Finally, on the Biogeosciences page that describes manuscript types, it states that Technical Notes "should be short (a few pages only)". However, we agree that the difference in open ocean and coastal pH trend magnitudes is interesting and worth pointing out in the context of the article. Thus, we have added a sentence near Line 114:

Line 114: Yet, at another Equatorial Pacific site (0 °N, 155 °W; Sutton et al., 2014), there is a similar [H+] trend to that of the Irminger Sea site because the initial pH differs. While we focus here on pH changes in the open ocean, pH changes also occur in coastal waters where they tend to be larger (Carstensen and Duarte, 2019). Recognizing that a change in pH represents a relative change in [H+], regardless of location, and examining long-term trends in both parameters should improve interpretation of chemical changes across ocean domains.

While pH is a useful notation for displaying wide ranges in [H+], hydrogen is the chemical element that organisms interact with in the environment. We think this is self-evident and haven't provided further comment on the matter as it would distract from the main point.

We agree that the comparison of proton concentration and saturation state changes with depth is useful in practice, but in this case would distract from the key point of the article.

We agree that proton budgets are useful for quantifying process contributions to local

acidification. However, this topic lies outside the scope of the article and would distract from the main point.

---

## Author Comment (AC2) · 15 Dec 2020

Thank you for providing constructive feedback on the manuscript. Below we address each comment from your review.

1. In this technical note, we cannot address all problems associated with interpreting pH changes. The manuscript is focused entirely on the common problem of misinterpreting or misrepresenting the meaning of pH changes, i.e., that a pH change represents a relative rather than an absolute change in [H+]. Addressing other problems would lengthen the paper and divert the reader from this sole objective. Furthermore, on the Biogeosciences page that describes manuscript types, it states that Technical Notes "should be short (a few pages only)". Thus, adding other concerns, such as this

analytical issue, is not feasible.

2. We agree that biological responses to ocean acidification are complex and require consideration of the full seawater chemistry, not just pH. However, we think this topic extends beyond the article scope. Additionally, unlike for all other carbonate system variables whose changes are absolute, the logarithmic scale of pH means that its changes are equivalent to relative changes in [H+]. We chose to emphasize this.

3. We have deleted this sentence.

4. In response to this comment, and feedback from a voluntary reviewer regarding the coastal ocean analogue to our open ocean examples, we have added a sentence near Line 114:

Yet, at another Equatorial Pacific site (0 °N, 155 °W; Sutton et al., 2014), there is a similar [H+] trend to that of the Irminger Sea site because the initial pH differs. While we focus here on pH changes in the open ocean, pH changes also occur in coastal waters where they tend to be larger (Carstensen and Duarte, 2019). Recognizing that a change in pH represents a relative change in [H+], regardless of location, and examining long-term trends in both parameters should improve interpretation of chemical changes across ocean domains.

---

## Author Comment (AC3) · 15 Dec 2020

Thank you for supporting publication of the manuscript and for providing constructive feedback.

Below we address each comment from your review in order of appearance.

The manuscript was submitted as a Technical Note rather than a Research Article because the concepts and content are not original. Instead, the manuscript explains an important aspect of pH change analysis and interpretation that is relevant to how the oceanographic community conceptualized ocean acidification. By providing concise explanations and clear illustrations of the key concept, the message can be delivered in a short, accessible format that is more likely to be read, leading to broader implementation of the recommendations.

We deleted the sentence on line 477 as it was unnecessary and potentially confusing.

The Figure 3 caption has been updated as follows: Shown are the decadal changes in (b) A-pH and (e) A-[H+] (nmol kg−1) relative to the 1950s as well as the surface ocean (c) pH and (f) [H+] (nmol kg−1) monthly anomalies relative to the annual mean at KEO and DPN during the 1950s (thin lines) and 2090s (thick lines). Global maps show the total change in (g) A-pH and (h) A-[H+] (nmol kg−1) between the 1950s and 2090s.

Referee #1 found the extensive referencing useful in guiding the reader to additional resources on related topics. Ocean pH changes are studied across a broad range of space and time scales and within numerous oceanographic subdisciplines. Each discipline contributes a unique perspective and applies different research tools. In effort to make the article relevant to a range of oceanographers considering pH changes, examples from different subdisciplines were used (i.e., work involving time-series, autonomous sensors, numerical modeling, and hydrography), requiring extensive citation of prior literature. Still, we have made an effort to reduce the total number of citations as well as redundant citation appearances throughout the text – shortening the reference list by 1.5 pages.